# Agronomic Investigation of Spray Dispersion of Metal-Based Nanoparticles on Sunflowers in Real-World Environments

**DOI:** 10.3390/plants12091789

**Published:** 2023-04-27

**Authors:** Dávid Ernst, Marek Kolenčík, Martin Šebesta, Ľuba Ďurišová, Hana Ďúranová, Samuel Kšiňan, Ramakanth Illa, Ivo Safarik, Ivan Černý, Gabriela Kratošová, Veronika Žitniak Čurná, Jana Ivanič Porhajašová, Mária Babošová, Huan Feng, Edmund Dobročka, Marek Bujdoš, Kristyna Zelena Pospiskova, Shadma Afzal, Nand K. Singh, Sasikumar Swamiappan, Elena Aydın

**Affiliations:** 1Institute of Agronomic Sciences, Faculty of Agrobiology and Food Resources, Slovak University of Agriculture in Nitra, Tr. A. Hlinku 2, 949 76 Nitra, Slovakia; 2Institute of Laboratory Research on Geomaterials, Faculty of Natural Sciences, Comenius University in Bratislava, Mlynská Dolina, Ilkovičova 6, 842 15 Bratislava, Slovakia; 3Institute of Plant and Environmental Sciences, Faculty of Agrobiology and Food Resources, Slovak University of Agriculture in Nitra, Tr. A. Hlinku 2, 949 76 Nitra, Slovakia; 4AgroBioTech Research Centre, Slovak University of Agriculture, Tr. A. Hlinku 2, 949 76 Nitra, Slovakia; 5Department of Chemistry, School of Advanced Sciences, VIT-AP University, Amaravati 522 237, Andra Pradesh, India; 6Department of Nanobiotechnology, Institute of Soil Biology and Biogeochemistry (ISBB), Biology Centre, Czech Academy of Sciences, Na Sadkach 7, 370 05 Ceske Budejovice, Czech Republic; 7Regional Centre of Advanced Technologies and Materials, Czech Advanced Technology and Research Institute, Palacky University, Slechtitelu 27, 783 71 Olomouc, Czech Republic; 8Nanotechnology Centre, CEET, VŠB Technical University of Ostrava, 17. listopadu 15/2172, 708 00 Ostrava, Czech Republic; 9Department of Earth and Environmental Studies, Montclair State University, 1 Normal Ave, Montclair, NJ 070 43, USA; 10Institute of Electrical Engineering, Slovak Academy of Sciences, Dúbravská cesta 9, 841 04 Bratislava, Slovakia; 11Department of Biotechnology, Motilal Nehru National Institute of Technology Allahabad, Prayagraj 211 004, Uttar Pradesh, India; 12Department of Chemistry, VIT University, Vellore 632 014, Tamil Nadu, India; 13Institute of Landscape Engineering, Faculty of Horticulture and Landscape Engineering, Slovak University of Agriculture in Nitra, Hospodárska 7, 949 76 Nitra, Slovakia

**Keywords:** foliar application, sunflower yield, nanofertilizers, seeds quality, gold, silica, zinc, iron oxide, plant physiology, insect biodiversity assessment

## Abstract

In environmental and agronomic settings, even minor imbalances can trigger a range of unpredicted responses. Despite the widespread use of metal-based nanoparticles (NPs) and new bio-nanofertilizers, their impact on crop production is absent in the literature. Therefore, our research is focused on the agronomic effect of spray application of gold nanoparticles anchored to SiO_2_ mesoporous silica (AuSi-NPs), zinc oxide nanoparticles (ZnO-NPs), and iron oxide nanoparticles (Fe_3_O_4_-NPs) on sunflowers under real-world environments. Our findings revealed that the biosynthetically prepared AuSi-NPs and ZnO-NPs were highly effective in enhancing sunflower seasonal physiology, e.g., the value of the NDVI index increased from 0.012 to 0.025 after AuSi-NPs application. The distribution of leaf trichomes improved and the grain yield increased from 2.47 t ha^−1^ to 3.29 t ha^−1^ after ZnO-NPs application. AuSi-NPs treatment resulted in a higher content of essential linoleic acid (54.37%) when compared to the NPs-free control (51.57%), which had a higher determined oleic acid. No NPs or residual translocated metals were detected in the fully ripe sunflower seeds, except for slightly higher silica content after the AuSi-NPs treatment. Additionally, AuSi-NPs and NPs-free control showed wide insect biodiversity while ZnO-NPs treatment had the lowest value of phosphorus as anti-nutrient. Contradictory but insignificant effect on physiology, yield, and insect biodiversity was observed in Fe_3_O_4_-NPs treatment. Therefore, further studies are needed to fully understand the long-term environmental and agricultural sustainability of NPs applications.

## 1. Introduction

Nanotechnologies enable new strategical opportunities in many industries involving agriculture by means of several novel and sometimes underexplored mechanisms [1,2,3]. Generally, nanoparticles (NPs) are defined as entities where at least 1 of the 3 dimensions does not exceed 100 nanometers. This dimension range provides a highly precise and sustainable tool for agriculture [3,4]. The dimensions, crystallinity, morphology, and high surface-to-volume ratio of NPs introduce unique chemical, physical, and biological properties, such as tunable reactivity and bioavailability, compared to their corresponding bulk or ionic counterparts [5]. Due to exclusive physiochemical properties, NPs can be effectively applied for stress mitigation strategies, e.g., heavy metal contamination of the soil [6,7].

The sunflower plant of the Asteraceae family is one of the most broadly cultivated oil bearing crops worldwide [8]. The quality of sunflower oil is highly dependent on the content of oleic and linoleic acids as the human body is unable to naturally produce these essential acids [9]. Additionally, the common sunflower is highly suitable for foliar application of NP due to their broad leaves acquisition [10]; diverse leave structures; surface morphology with stomata; trichomes diversity [11,12]; and suitable ability to take up, transport, transform, or accumulate various metals to different parts of the plants [13]. Additionally, the sunflower cultivation environment can be adequately utilized for an ecological insect diversity estimation [14,15] in which the metallic NPs may be able to penetrate with varying degrees of harmful effects on terrestrial organisms [16].

Newly applied NPs, present in low range-concentrations, have the advantage of gradual nutrient release and more targeted delivery, which is in contrast to commercially available fertilizers [17]. Additionally, spray dispersion research is increasingly focused on the green synthesis of metal-based nanoparticles. Bio-synthetically prepared metal and metal oxide NPs can have various positive effects on crop production, including improving environmental stress tolerance against drought [3,18,19]. Appropriate physiological response of plants could be accelerated synergically due to combination of metal-related nanofertilizes with organic residual compounds from biological synthesis [3].

Nanoparticles primarily enter plant bodies through their leaf, stem, and root systems. Liu and Lal [17] categorized Fe, a Zn-based NP, into micronutrient groups of fertilizers with relative essential responses in plants. However, there is limited information available on the combined effect of Au-NPs and SiO_2_ composite materials. Elemental zinc is a plant micronutrient required for proper metabolism and acts as an essential part of more than 300 enzymes. It supports photosynthesis functions, regulates auxin and hydrocarbon metabolism, encourages cell membrane integrity, DNA transcription, and performs roles in other biological processes [20]. Iron deficiency results in leaf chlorosis [21], negatively influences the morphology and physiology of roots in the soil environment [22], and is an integral part of proteins and transporters [23]. Silica is not essential for plants, but it is bioavailable in the form of soluble silicic acid. Higher content of silica showed enhancement of plant vitality, exogenous stress responses, and provided mechanical strength to the arial parts of plant [24]. Elemental gold does not classify as an essential element. Cations of Au^3+^ can inhibit plant growth and development, and change plant physiology, biochemistry, and molecular structures to a certain level [25]. In this context, in their review, Siddiqi and Husen [25] present the role of Au NPs with several positive effects on plants which were observed at concentration corresponding to 10 μg mL^−1^. Our previous study has shown that ZnO-NP application positively correlated with sunflower head diameter and weight of dry seeds [26]. Spray application of Fe_3_O_4_-NPs increased dry and fresh weights of *Nicotiana benthamiana* Domin plant [27]. Similar effects on plant height, stem diameter, and seed yield were observed for SiO_2_-NPs application to *Vetiveria zizanioides* L. Nash [28] and for Au-NPs to *Brassica juncea* L. Czern. [29]. Content of sunflower oil was higher after application of TiO_2_-NPs. ZnO-NPs were associated with improved sunflower or foxtail millet physiology despite the seasonal climatic variability [26,30]. Moreover, the quality of sunflower fully ripe seeds can be affected by mineral nutrients, such as copper, phosphorus, or zinc [9,31]. However, the mechanisms of potential uptake and translocation of NPs and their metal derived residues to fully ripe seeds or impact on the profile of macro, micronutrients, and trace elements are still unknown.

The use of photocatalytically active NPs is an attractive platform for enhancing plant physiological reactions [32]. Grimme et al. [33] hypothesized that metal NPs can accelerate the energy transfer leading to an increase in plant photosynthesis reactions and resulting in higher quantitative parameters and yield. ZnO-NPs belong to a II-VI semiconductor [34] and mesoporous silica (SiO_2_) provides uniform pore size architecture with huge surface-to-volume ratio, allowing for anchoring of Au-NPs with their photocatalytic properties that can be activated by quantum yield effect. Gold NPs induced by sunlight wavelengths generate enable easier electron transfer which may help in photosynthesis [35]. In addition to its semiconductor properties, iron oxide (Fe_3_O_4_) also has either photocatalytic or superparamagnetic properties [36].

Current research of NPs spray deposition to plants focuses mainly to laboratory studies or greenhouse conditions [37,38]. Field experiments and applications in agronomy are still mostly absent. Due to aforementioned facts, our complex study investigated (i) the agronomical aspects of sunflower physiology, yield related parameters, and surface leaf morphology reactions; (ii) the final quality assessment with potential translocation of NPs or their metal residues, ratio of essential acids in sunflower oil, and anti-nutrients content; and (iii) agroecological impact on insect biodiversity based on the determination of epigeic insect groups.

## 2. Materials and Methods

### 2.1. Origin and Characterization of Sprayed Nanoparticles Applied on Sunflower

The foliar application of colloidal solution of zinc oxide NPs to sunflower had concentration of 15.99 mg L^−1^, and the NPs were commercially acquired from Sigma-Aldrich (Saint Louis, MO, USA). Detailed characterization of the NPs morphology, crystallinity, dimensions, and structure parameters was published by Kolenčík et al. [30] and colloidal characteristics, such as hydrodynamic diameter and zeta potential, are present in the work of Kolenčík et al. [39].

Another treatment used the mesoporous algae cells of *Mallomonas kalinae* sp. nov. with silica-based composition (SiO_2_) with anchored well-crystalline gold NPs with three types of morphology and approximate dimension of 10 nm. Special photocatalytic properties of this composite material have been already confirmed by observing nerve agent degradation [40]. For our experimental purposes, we applied 0.1 mg L^−1^ of Au-NPs with 10 mg L^−1^ of SiO_2_ corresponding to mesoporous algae cells of *Mallomonas kalinae* sp. nov.

Iron oxide NPs (Fe_3_O_4_-NPs) were prepared according to Domingo et al. [41] and we used it at the concentration of 76 mg L^−1^. Citrate buffer was used for colloidal stabilization of Fe_3_O_4_-NP. Crystal structure characteristics and dimension calculation was carried out by X-ray diffraction analysis (XRD). XRD measurement was performed by BRUKER X8 DISCOVER diffractometer (Bruker, MA, USA) at the following conditions: 12 kW (40 kV, 300 mA) with Cu-anode. Gained data were used for calculation of unit cell parameters with TOPAS 3.0 software (Bruker, MA, USA) (Appendix A). Fe_3_O_4_-NP*s* morphological interpretation was studied by transmission electron microscopy (TEM) on JEOL JEM 2100 (JEOL, Tokyo, Japan) at 200 kV of accelerating voltage. Chemical composition of samples was analyzed by energy dispersive X-ray spectroscopy (EDS), spectra were acquired by scanning electron microscope (SEM) using Thermo Noran System 7 (Thermo Scientific, Waltham, MA, USA) with Si (Li) detector, accelerating voltage was 15 kV and acquisition time was 300 s.

### 2.2. Plant Material

Common sunflower (*Helianthus annuus* L.) hybrid SY Neostar (Syngenta, Basel, Switzerland) was used in our field experiment. This hybrid belongs to the two-line imidazoline resistant hybrid preferentially utilizable for ClearField Plus^®^ system of production. SY Neostar hybrid has a short to medium height, medium to early growth and development, with a wide adaptability against inadequate environmental and growth conditions, and no specific agricultural requirements. Additionally, this hybrid contains around 47% of oil on average, shows high resistance to *Sclerotinia sclerotiorum* Lib. de Bary, *Diaporte helianthi* Munt.-Cvetk., Mihaljč., and M. Petrov, and is tolerant to *Plasmopara halstedii* Farl. Berl. and De Toni.

### 2.3. Experimental Location Description

The field study was carried out in the experimental fields of Slovak University of Agriculture in Nitra (SUA in Nitra), at Dolná Malanta close to Nitra (48°19′25.41″ N 18°09′2.89″ E), Slovak Republic, central Europe. The location is situated in the north-east part of Panonian Lowland, south of the Tribeč mountains at an elevation of 250 m above sea level. Basic petrological composition of the rock corresponds to granite-based and Mesozoic carbonate rocks, Neogene, Quarter development with eluvial sediments [42]. Soil was classified as silt loam haplic Luvisol [43]. X-ray powdered diffraction analysis (XRD) of soil was conducted by diffractometer PW1710 (Philips, Amsterdam, The Netherlands). The soil minerals composition according to the X-ray diffractogram corresponds to the quartz, muscovite, and anorthite (Appendix A). The research locality is situated in a typical maize crop region with intensive soil farming practices and the current sunflower was cultivated in 7-plot crop rotation.

### 2.4. Climatic Seasonal Variations

Variability of the precipitation accumulation and average daily temperature during vegetation season in 2019 was monitored at meteorological station of SUA in Nitra (Figure 1).

### 2.5. Field Experiment

The field experiment was conducted by a method of four randomly organized treatments in perpendicularly selected blocks according to Kolenčík et al. [26]. Each treatment consisted of three replications. The plot area of one replication was 60 m^2^.

The field was deeply ploughed with a Zetor tractor 6211 (Zetor Tractors, a.s., Brno, Czech Republic) in autumn 2018. The forecrop was winter wheat (*Triticum aestivum* L.). Soil characteristics were analyzed before sowing of sunflower in spring 2019 and are shown in Table 1.

Urea fertilizer was applied by machinery, i.e., by tractor, during pre-sowing soil cultivation using a Ferti fertilizer applicator (Agromehanica, Boljerac, Serbia). The fertilizer was applied on the basis of agrochemical analyses before sunflower sowing at a dose of 215 kg ha^−1^.

Soil sampling for nitrogen fertilization was carried out during pre-sowing soil cultivation in spring 2019. The results of soil sampling for nitrogen fertilization are presented in Table 2.

Sunflower was sown in the rows to a depth of 60 mm in 220 mm seed distance and 700 mm inter-row spacing using a Monosem NG Plus 3 planter (Monosem, Largeasse, France) [44]. Then, 4 L ha^−1^ of Wing^®^ herbicide (BASF, Ludwigshafen am Rhein, Germany) was applied on the third day after sowing. All treatments, including the control, were sprayed with standard herbicide and fungicide by AGT 865T/S sprayer (Agromehanica, Boljerac, Serbia).

Colloidal solutions of NPs with a concentration of 15.99 mg L^−1^ ZnO, 76 mg L^−1^ Fe_3_O_4_ and 0.1 mg L^−1^ Au anchored in mesoporous SiO_2_ at a concentration of 10 mg L^−1^ were prepared in water and foliarly applied onto the plants using a hand-held sprayer (Mythos Di Martino, Mussolente, Italy). Application was performed early in the morning in windless conditions until the sunflower leaves were completely wet. The first spray application was made at day 40 at the leaf development stage, and the second spray at day 80 at the stem elongation stage at flower bud formation (Figure 2). In control treatment, only water was used for preparation. Additionally, the same water was used for colloidal NPs solutions and subsequently applied to the sunflower leaves [26].

### 2.6. Microscopical Analysis of Trichomes and Stomata of Sunflower Leaves after Various Nanoparticles Treatments

Sunflower leaves were collected for surface structure investigation one week after the second spray deposition at a stage of flower bud formation. For precise determination of changes of surface structures and morphology analysis, the visualization by scanning electron microscopy with Quanta 450 (FEG, FEI, Hillsboro, OR, USA) was used according to studies conducted by Aschenbrenner and Horakh [45]. Trichome distribution and their basic features were approximately estimated from leaf area of 1 mm^2^ (=1,000,000 µm^2^). Samples for stomata observation were prepared by microrelief (replica) method. Sunflower leaves were covered with colorless nail polish diluted with acetone. Replicas of stomata made with duct tape were placed on a glass slide [46]. Stomata were observed by a light microscope Olympus BX 41, only on the abaxial side of the leaf. Photographs of stomata were made by Olympus E 520. Overall, four repetitions were performed per treatment (including control). In each repetition, 10 random photographs of leaf epidermis were made. The following list contains parameters that were evaluated: length and width of stomata, length and width of stomatal pores, and number of stomata per 100,000 µm^2^.

### 2.7. Evaluation of Quantitative and Nutritive Parameters of Sunflower

After reaching full seed maturity, sunflowers were harvested by small pot combine harvester Claas Dominator 38 (CLAAS GmbH & Co. KGaA, Harsewinkel, Germany). The seed moisture level was analyzed by He Lite (Pfeuffer GmbH, Kitzingen, Germany). Yield was calculated in tons per hectare (t ha^−1^). Parameters, such as manual count of plants with heads (pcs), head diameter (mm) (Texi 4007 laboratory equipment; Texi GmbH, Berlin, Germany), head weight (g), and thousand seed weight (g) (TSW) (Kern PCB3500-2 lab scale; KERN & Sohn GmbH, Balingen, Germany and Numirex seed counter MEZOS spol. s r.o., Hradec Králové, Czech Republic), were examined for each treatment.

### 2.8. Analysis of the Course of Sunflower Physiology Based on Selected Physiological Indices

PlantPens NDVI 300 and PRI 200 (Photon Systems Instruments, Brno, Czech Republic) devices were used to evaluate the normalized difference vegetation index (NDVI) and photochemical reflectance index (PRI). The principle of the analysis is based on the absorption of reflected wavelengths from the leaf surface (at 740 nm and 660 nm for NDVI, and at 531 and 570 nm for PRI, respectively). The indices for NDVI and PRI were calculated according to the following Equations (1) and (2):NDVI = (R_740_ − R_660_)/(R_740_ + R_660_)(1)
PRI = (R531 − R570)/(R531 + R570)(2)
where R_531_, R_570_, R_660_, and R_740_ correspond to the reflecting wavelengths from the leaf surface, respectively. The above altered non-destructive methods were performed on the same day, on the same plants, on fully developed leaves, at the same developmental stage between 11.00 a.m. to 1.00 p.m. To include leaf heterogeneity, minimum of 10 point perpendicular-oriented measurements of leaf per index were made according to Gamon et al. [47].

The EasIR-4 thermo-camera (Bibus AG, Fehraltorf, Switzerland) was used to analyze the stomatal conductance index (Ig) and the crop water stress index (CWSI) according to Jones et al. [48], where leaf temperature parameters (T^leaf^), dry (T^dry^), and wet (T^wet^) leaf surface and atmospheric moisture were included. Temperature images were taken diagonally on sunflower leaves from a distance of 2 m, at a height of 1.5 m, and at a resolution of 20.6° × 15.5° in auto-focus mode. Ig and CWSI were calculated according to Equations (3) and (4):Ig = (T^dry^ − T^leaf^)/(T^leaf^ − T^wet^)(3)
CWSI = (T^leaf^ − T^wet^)/(T^dry^ − T^wet^)(4)

### 2.9. Sunflower Quality Assessment with Potential Translocation of Nanoparticles, Evaluation of Essential Acids in Sunflower Oil, and Anti-Nutrients Content

For quantifying the amount of sunflower oil, Soxhlet extraction was used [49]. Analysis and profile of free fatty acids was carried out with a few sequential steps. First, the separation based on carbon numbers and saturation levels were conducted. Second, extraction was utilized on the shortest carbon chains with the lowest boiling points. Then, individual free fatty acids were detected by flame ionization detector (FID) integrated to the gas chromatograph Agilent 6890A GC (Agilent Technologies, Santa Clara, CA, USA). Profile of free fatty acids were evaluated as percentage of crude fat and other more complex methodological specifications are provided by Kolláthová et al. [50].

Translocation and accumulation of applied NPs (zinc, gold–silica, and iron) and their potential transfer residues to kernel and hulls in full ripe seeds were analyzed using ICP-MS (Thermo Scientific iCap-Q, Bremen, Germany) according to methodologies published by several authors [26,51,52,53]. The calcium, phosphorus, and magnesium corresponding to mineral anti-nutrients of sunflower seeds were detected by ICP-OES (Varian Vista MPX, Mulgrave, Victoria, Australia) with yttrium as an internal standard. Potassium and manganese were detected by F-AAS (Perkin-Elmer Model 1100, Waltham, MA, USA). The colorimetric Kjeldahl method was used to define the total content of nitrogen and sulfur. All aforementioned elements were analyzed after digestion procedures of 0.15–0.30 g sample with Anton Paar Multiwave 3000 microwave digestion system (Graz, Austria) in PTFE pressure vessel. The mixture of concentrated HNO_3_ and H_2_O_2_ at the pressure of 60 bars was used for dilution.

### 2.10. Examination of Nanoparticles Spray Application on Sunflowers on the Agro-Ecological Biodiversity Refection

The insects with predominant epigeic components were collected by the earth traps method. The method was based on 1 L jars, which were placed directly into soil and covered for protection against rainfall and rodents in all NPs-treatments and control (NPs-free). Periodically, in monthly intervals, jars were checked and refilled up with mixture of 5% of formaldehyde and 1/3 of fixative solution. For biological material determination, the typical insect features, especially impacted to Carabidae family were observed and systematically classified according to the work of Pokorný [54]. The agro-ecological biodiversity was estimated in accordance with the presents epigeic population (abundance) and determination of *Coleoptera* and *Carabidae* obtained families.

### 2.11. Statistical Analysis

The observed data from trichomes and stomata analysis were statistically examined with a Tukey HSD test at α = 0.01 significance. The statistical analysis for all other treatments was performed with Statistica 10 software (StatSoft, Inc., Tulsa, OK, USA) [55]. Prior to the evaluation of the multifactorial analysis of variance (ANOVA), the normality of experimental data was tested at α = 0.05 and α = 0.01 significance by the Student t-test, Shapiro–Wilk test for trials, and Fisher’s least significant difference (LSD).

## 3. Results

### 3.1. Characterization of Spray-Applied Nanoparticles

Commercially available ZnO-NPs showed a dominantly spherical shape, was less frequently a hexagonal or cuboidal shape (Figure 3), and had the high crystalline nature with corresponding wurtzite type structure with an average dimension around 17.3 ± 0.1 nm. Holišová et al. [40] biosynthetically prepared gold NPs with 10 nm dimension with spherical, rarely trigonal morphology (Figure 3b), which was anchored directly to mesoporous biosilica (algae *Mallomonas kalinae* sp. nov.). Due to the spray deposition on the sunflower leaves, the “composite” AuSi-NPs with residual solution did not promote pure NPs-properties and responses. In the case of synthetically prepared Fe_3_O_4_-NPs with citrate buffer, the shape of Fe_3_O_4_-NPs was mostly spherical or pseudo-octahedral and particle size distribution ranged around 5.4 ± 0.2 nm (Appendix A). Chemical composition verification through EDS analysis (Figure 3c) determined elements of Fe, O, and crystal structure corresponding to the magnetite mineral (Appendix A). The other structure-related parameters are given in Appendix A.

### 3.2. Evaluation of Leave Surface Structures—Trichomes and Stomata after Spraying with Nanoparticles

The total number of trichomes increased on the adaxial side of the leaf in all treatments when compared to the control (Table 3). The highest number of non-glandular trichomes (NGT) was recorded in the treatment with AuSi-NPs application, but they were also the smallest. In the case of linear glandular trichomes (LGT), their number increased only in the treatment with ZnO-NPs, while the number of trichomes in other treatments (with Fe_3_O_4_-NPs and AuSi-NPs) was lower when compared to the control. The length of LGT trichomes on the adaxial side of the leaf was greatest in the control, while shorter LGT trichomes occurred in all treatments after NPs application (Appendix A). A statistically significant difference in the length of this type of trichomes was recorded in the treatment Fe_3_O_4_-NPs. All types of sunflower trichomes are shown in Figure 4.

The total number of trichomes on the abaxial side of the leaf in treatments with AuSi-NPs and ZnO-NPs was higher than control, while in the treatment by Fe_3_O_4_-NPs, the density of trichomes was lower. The same trend was observed in NGT trichomes. The length of Fe_3_O_4_-NPs NGT trichomes was significantly larger than control. In the case of the number of LGT trichomes on the abaxial side, a higher number of trichomes was recorded only in the AuSi-NPs treatment, while in the Fe_3_O_4_ and ZnO-NPs, their numbers were lower than control. No statistically significant difference in the length of this type of trichomes was recorded. We observed that CGT trichomes only occurred on the abaxial side of sunflower leaf. The smallest number of this type of trichomes was recorded after ZnO-NPs application, but they were significantly larger (57.63 µm) when compared to the control and AuSi-NPs treatment.

The longest stomata were recorded in the control treatment (30.10 µm) and the shortest was in the AuSi-NPs application treatment (28.67 µm) (Appendix A). The stomata length in AuSi-NPs treatment was also significantly different when compared to control. The widest stomata were also found in control treatment (18.99 µm). The narrowest stomata were recorded in Fe_3_O_4_-NPs treatment (17.90 µm). Statistically significant difference was recorded between Fe_3_O_4_-NPs and AuSi-NPs, and between Fe_3_O_4_-NPs and control treatment. The longest stomatal pores were recorded in control treatment (18.30 µm) and the shortest in Fe_3_O_4_-NPs application treatment (17.69 µm). Statistically significant difference was between Fe_3_O_4_-NPs and AuSi-NPs treatments. The widest stomatal pores were recorded in control treatment (3.19 µm). The narrowest stomatal pores were in Fe_3_O_4_-NPs treatment (2.82 µm). A significant difference was recorded between control treatment and the group of other treatments after NPs application. The mean number of stomata per 100,000 µm^2^ was highest in control treatment (25.03) and lowest in Fe_3_O_4_-NPs treatment (20.08). There was a significant difference between Fe_3_O_4_-NPs treatment and two other treatments (control and ZnO-NPs). Numbers of stomata per 1 mm^2^ are listed in Appendix A.

### 3.3. Effects of Nanoparticles on the Sunflower’s Quantitative and Nutritional Parameters

There were no observed differences between the number of plants and number of heads per hectare for treatments with NPs application when compared to control (Table 4). The most significant differences were shown at AuSi-NPs and ZnO-NPs treatments in quantitative parameters, such as head diameter, weight of dry seed head, and weight of thousand seeds (TSW) compared to Fe_3_O_4_-NPs and control treatment. Additionally, ZnO-NPs provided significantly higher grain yield when compared to control. The least effective treatment for selected parameters was Fe_3_O_4_-NPs, except the slightly positive response in weight of dry seed head. Application of NPs did not have effect on the content of oil in sunflower seeds.

### 3.4. Sunflower Full Ripe Seed Quality

Sunflower oil contained the main essential fatty acids, such as oleic, linoleic, palmitic, and stearic acids; however, significant differences were observed only for oleic and linoleic acids. While the content of oleic acid significantly decreased in all treatments with NPs when compared to control, the content of linoleic acid significantly increased after NPs application. Other analyzed compounds had relatively insignificant contents (Table 5).

Higher content of elements after NPs spray deposition to sunflower was partly observed only for AuSi-NPs treatment, where relatively higher content of silica was observed when compared to control (Table 6). Gold content was not detected in any of the samples (detection limit < 0.002 mg L^−1^). The ZnO-NPs and Fe_3_O_4_-NPs treatments did not show any transport or accumulation of NPs or their residues into full ripe seeds. In contrast, the control treatment indicated slightly higher concentration of zinc and iron (Table 6). Mineral nutrients indicate differences in control with its higher content at the P level against ZnO NPs variant, while Mg, Ca, and K have lower value. In AuSi-NPs treatment, the concentration of Fe, Si, and K was slightly higher in contradiction to other treatments, including control. Application of Fe_3_O_4_-NPs did not significantly affect the content of analyzed mineral nutrients.

### 3.5. Sunflower Physiological Response to Different Nanoparticles Treatments

The total seasonal average physiological values on the NDVI, PRI, and IG, which correspond to photosynthesis activity, chlorophyll content, and stomata conductance, were significantly different in the case of AuSi-NPs and ZnO-NPs treatments when compared to control. Additionally, positive response was shown at Ig index for application of Fe_3_O_4_-NPs. The lowest value of CWSI, which agrees with the most effective sunflower water management, was again associated with AuSi-NPs and ZnO-NPs treatments when compared to NPs-free control (Table 7).

The effect of nanoparticles treatments on different physiological indexes during the vegetation season in 2019 is shown in Figure 5a–d. After the first application of AuSi-NPs, significant trend of positive physiological reactions at PRI, IG, and CWSI were observed. This enhanced physiological tendency with slight exceptions continued after second NPs-application to seed development and ripening for all physiological indexes when compared with NPs-free control. Almost analogical physiological responses followed ZnO-NPs-treatment, which indeed were achieved for NDVI, PRI, and CWSI indexes after second application to ripening stage. In the case of Fe_3_O_4_-NPs treatment, there was evidently huge uncertainty during seasonal physiological responses, but results of all indexes indicate that after second application to harvesting stage, there was a delay in inappropriate senescence and aging of sunflower.

### 3.6. Impact of Spray Nanoparticles Application to Agro-Ecological Diversity Based on Epigeic Groups under Field Conditions

Our agro-ecological study showed that under NPs treatments and NPs-free control, 4738 individuals of various epigeic fauna were collected. Found individuals belonged to 20 taxonomical groups spatially unequally distributed within experimental treatments. The most abundant groups were Coleoptera (44.38%), Collembola (19.93%), and Acarina (11.67%). The subdominant group was represented by Araneida (4.49%), Opilionida (4.29%), and Orthoptera (2.76%). Nondetermined stage of Larvae (1.77%) and Diptera (1.19%) occurred in the minority group. Abundance and dominance of other 11 epigeic groups did not exceed more than 1%. Spatial-temporal distribution of trapped epigeic groups had generally decreasing tendency. From a biodiversity point of view, the most convenient life conditions of epigeic groups were found in treatments in the following order: AuSi-NPs treatment > NPs-free control > Fe_3_O_4_ NPs treatment > ZnO-NPs treatment (Table 8). Most unexpectedly, ZnO-NPs treatment offered inappropriate conditions for life cycle of populations.

An analogical trend was observed for the Coleoptera family in the case of treatments with AuSi-NPs application and NPs-free control in comparison to Fe_3_O_4_-NPs and ZnO-NPs (Appendix A). The whole Coleoptera family was the most dominant with 11 groups, whereas Carabidae reached the most quantitative dominance with 93.44%. Recedent occurrence was also observed for Anthicidae, Dermestidae, and Elateridae families. The abundance of other families was lower than 1%.

Differences in species abundance between the AuSi-NPs treatment, control, and Fe_3_O_4_-NPs and ZnO-NPs treatments were also confirmed for the Carabidae species including either zoophagous or macropterous species (Appendix A). Herein, *Harpalus rufipes* Degeer was the most eudominating species with maximal appearance in late summer and autumn period. *Brachinus crepitans* L. (9.92%) and *Poecilus cupreus* L. (4.37%) were also comparatively abundant. The representation of other species was less than 1% of the whole population (Appendix A).

## 4. Discussion

### 4.1. Changes of Leaf Surface Structures—Trichomes and Stomata after Nanoparticles Application

According to several studies [1,39], spray deposition of inorganic NPs to leaf surface and other crop parts can induce any type of effects (positive, negative, or no effect). This could be related to the properties of NPs, their potential essentiality, and activity of ambient solutions. In our experimental platform, sunflowers treated by (i) commercially available micronutrient-based zinc dioxide nanoparticles (Figure 3a) dispersed in aqueous solution, (ii) bionanotechnological prepared gold nanoparticles fixed within mesoporous diatomite (AuSi treatment) with residual algae-cultivation medium (Figure 3b) and iron oxide (Fe_3_O_4_-NPs) formed and stabilized in citrate buffer (Figure 3c,d). Liu and Lal [17] categorized gold and silica into postulates with relatively unclear mechanisms to plants; however, the gold silica biocomposite affiliates with all benefits required within the new generation of nanobiofertilizers [56].

One week after the second application, no correlation was observed between trichomes and stomata on the abaxial and adaxial side of leaf structures in both NPs-treated and control treatments (Table 3 and Table 4). In the case of Fe_3_O_4_ NPs and ZnO NPs, abaxial side of leaves provided less distribution of trichomes than adaxial side of leaves, which is in contrast with the principles related to *Heliantheae* tribe practice [57]. Concentration of trichomes on the adaxial side of leaves in NPs-free control was reduced in contrast with increasing number of them in the NPs treatments. The highest number of statistically significant differences was observed in ZnO-NPs treatment, but the number of trichomes did still not reach the standard values presented by Read et al. [58]. A similar trend was observed with stomata (Appendix A), where the highest number of them was found in ZnO-NPs treatment and NPs-free control in comparison with Fe_3_O_4_ a AuSi-NPs treatments. Askary et al. [59] observed density changes of all trichomes types on the sunflower leaf surface, namely of non-glandular trichomes (NGTs), linear glandular trichomes (LGTs) and capitate glandular trichomes (CGTs) after application of Fe_3_O_4_-NPs. Additionally, authors described a trend of hair density reduction on the leaf surface of *Mentha piperita* L. but in our experiments the distribution of trichomes had increasing tendency. This is most likely a kind of response to slightly higher iron concentration and monodispersed range, smaller size of nanoparticles or fluctuations in seasonal field conditions (Figure 1). An increase in hair density was also declared by Da Costa et al. [60] after application of CuO-NPs. Authors hypothesized that the morphological changes, such as intensified size and number of trichomes and decreasing size of stomata were associated with Cu-induced osmotic stress. The same mechanisms probably occurred in our Fe_3_O_4_-NPs and ZnO-NPs treatments with sunflowers because macronutrient-related nanoparticles have almost similar solubility, essentiality, and potential bioavailability [17]. In the case of impact of AuSi-NPs on the leaf structures, it could be considered unpredictable stress factors, but now, detailed information how to react is widely absent in the academic literature.

Each type of trichomes performs a distinctive role for plant protection and it performs several functions, such as cover intake, absorption, and accumulation with various substances, including nanoparticles [1,45]. The most abundant trichomes with shortest size—NGTs—were found on adaxial side of leaves in AuSi-NPs treatment (Table 3). Silica was most likely integrated and accumulated directly to the basal part of leaf trichomes with phytolites formation [61]. In general, elemental silica does not belong to essential nutrients, but several works pointed to its higher plant-tolerant function against abiotic and biotic stress [61], ability to reduce accumulation of reactive oxygen species (ROS), membrane lipid peroxidation, access inhibition to toxicants, and enhanced plant defense to pathogens [62].

The abaxial side of leaves was in good agreement with the size and numbers of trichomes in Fe_3_O_4_-NPs treatment where the smallest hair density with its size was probably induced by NPs stress reactions. This could be related to the size of Fe_3_O_4_-NPs, which is around 5 nm or less (Figure 3c,d, Appendix A), which could distinctly coincide with cluster systems [63]. Additionally, it underlines the unique effect to LGTs on abaxial side of leaves with supposed penetration, short way transport, and inhibition of LGTs distribution [13,64]. Relative insignificant discrepancies were observed even at trichome width of NGTs and LGTs (Table 3), but the finding agrees with unified range of sizes published by Read et al. [58].

Furthermore, it is possible that not only the NPs themselves, but also residual components of the cultivation medium used in the biosynthesis of AuNPs may have had a slight impact on sunflower leaf structures. Some of these components may have acted as growth-promoting nutrients [40]. Similarly, the ferrofluid containing Fe_3_O_4_-NPs used in our study also contained residual citrate, which may have affected the absorption of NPs or the surface structures of the leaves due to the role of citrate in metabolic pathways in sunflower leaves [12].

### 4.2. Effects of Nanoparticles on the Sunflower Quantitative and Nutritional Parameters

Spray-dispersed application of NPs did not statistically affect the changes in the number of plants and the number of heads per hectare in comparison with NPs-free control (Table 4). Consistent with our previous studies [26,30,39], the current results further support the concept that low concentrations of NPs are effective for enhancing the sunflower growth. In the case of ZnO-NPs and Fe_3_O_4_-NPs, negative effects, such as reduced plant growth and stress symptoms, have been observed in response to concentrations exceeding 1000 mg.L^−1^ of ZnO-NPs or the application of agrochemicals in macro-sized and soluble ionic forms [65,66]. In this context, positive effect on quantitative parameters could reflect soluble-species of Fe and Zn with organic compounds, e.g., chitosan [67].

The positive results on head diameter, weight of dry seed head, and TSW quantitative parameters at ZnO-NPs and AuSi-NPs treatments were similar to ones in our previous study [26]. However, the most unexpected results were found for AuSi-NPs treatment in case of head diameter and weight of dry seed head, most probably for the reasons involving suitable stress physiological reactions often associated with SiO_2_ nanomaterials [28] or Si-based composites easy serving delivery of NPs [68]. In laboratory conditions, Arora et al. [29] observed an increase in quantitative parameters and yield of *Brassica juncea*, including plant height, stem diameter, number of branches, number of pods, and seed yield, following spray application. In contrast, also in laboratory conditions, Tomaszewska-Sowa et al. [69] found that the effect of Au-NPs in concentrations 0, 50, and 100 mg L^−1^ caused reduction in the shoot length and weight, as compared to control. It is also known that foliar application of Si-NPs significantly can improve the growth of wheat [70]. Seleiman et al. [19] stated that foliar applications of ZnO and SiO_2_ nanoparticles significantly enhanced number of tubers, tuber fresh and dry weight, tuber yield per plant, tuber yield per m^2^ with concentration at 50 and 100 mg L^−1^ in the case of SiO_2_-NPs, and 25 and 50 mg L^−1^ for ZnO-NPs of potato plants in water deficit treatments.

On the other hand, the weight of thousand seeds and content of oil at AuSi-NPs treatment were almost analogical with other treatments (e.g., Fe_3_O_4_-NPs or NPs-free control); however, agronomical interest increase was reached after calculation of pure oils per tons. In perception of yield related parameters, ZnO-NPs treatment was the most effective as it was proved earlier by foxtail millet [30] or common sunflower [26]. The Fe_3_O_4_-NPs treatment was not effective, except for the weight of dry seed head. It is relatively surprising finding because several studies published various beneficial effects, e.g., Cai et al. [27] described promoted plant growth and elicit protective response against plant virus to *Nicotiana benthamiana* (Domin) after application of Fe_3_O_4_-NPs at higher concentration at 100 μg L^−1^. Furthermore, Shahrekizad et al. [71] observed intensification of the arial organ dry mass or plant height with higher content of iron in the case of sunflower with spray deposition of Fe_3_O_4_-NPs-EDTA. Mahmoud et al. [72] stated that the foliar application of FeNPs significantly enhanced the number of branches, and fresh and dry weight of faba bean plants. The number of branches per plant was significantly higher due to the use of Fe-NPs compared to the control treatment. Similar results were observed in the fresh and dry weight of shoots.

### 4.3. Evaluation of Final Sunflower Seed Quality in the Context of Translocation, Fatty Acids Profiles, and Mineral Nutrients

Full ripe sunflower seeds generally vary in content of crude proteins ~ 20–40%, lipids ~ 47–65% [73], moisture ~ 2.5–6.30%, and ash ~ 2.7–4.9% [9]. Our study showed an increase in oil content from 47.5 ± 3.28 to 52.24 ± 1.27% in seeds treated with NPs (Table 4), which is within the range for high-oleic sunflower ~ 37.5–54% [9]. Similar results were observed in a previous study conducted at the same experimental locality with the same cultivar and agronomical management using low concentrations of ZnO-NPs and TiO_2_-NPs [26]. The quality of sunflower oil is assessed based on fatty acid profiles, its physico-chemical properties, antioxidant potential, and content of others compounds [74] which are nowadays widely utilized for the food or energy industry—biodiesel production [75]. Our results showed a favorable ratio of linoleic to oleic acids for the food industry (Table 5), as oleic acids have been linked to cholesterol reduction, prevention of heart disease and cancer, improvement of brain function, and reduction in autoimmune and inflammatory diseases [76,77]. However, the treatment without NPs showed the statistically most significant value of oleic acid content, while linoleic acid content increased in all NPs-treated samples (Table 5). The concentration of other fatty acids and compounds was not significantly different among the treatments (Table 5).

There were no significant differences observed in the translocation of Zn, Fe, and Au or their residuals in the treatments with NPs, which could potentially negatively influence the quality of final products (Table 6) and this was congruent with our previous study on lentil [39]. However, relatively surprising results were observed for Fe_3_O_4_-NPs, where we expected higher translocation because of higher applied concentration and their smaller size ~5.5 nm (Appendix A; Figure 3c,d). Oppositely, dramatic increase in iron aerial biomass content with Fe_3_O_4_-NPs-EDTA was published by Shahrekizad et al. [71].

Slightly higher but significant content of silica was found in full ripe sunflower seeds treated with AuSi-NPs compared to NP-free control. We believe that biogenic silica from *Mallomonas kalinae* sp. nov. [40] may be more bioavailable than more mechanically, chemically stable non-biogenic forms of SiO_2_ [78]. A promising positive effect of silica on sunflower was observed for non-glandular trichomes (NGT) on leaf surfaces in AuSi-NPs treatment as trichome had generally higher silica content (Appendix A). Therefore, our results confirmed a kind of silica accessibility process, which is important for knowledge on its biogeochemical cycle. On the contrary, it could be also a kind of reaction to seasonal dynamics or others undisputable factors.

The mineral nutrient content of dehulled sunflower seeds is typically between 3% and 4%, which is consistent with our study [73]. Phosphorus content is particularly important to consider due to its potential antinutrient effects in humans, primarily in the form of phytic acid [31]. Other soluble macronutrients, such as Ca, Mg, and K are readily eliminated by the human body after digestion [79]. Our study found that the NPs-free control treatment had a statistically higher content of phosphorus. Additionally, negative physiological responses were observed throughout the sunflower life cycle, particularly during the ripening and seed formation stages (Figure 5a–d) (Table 7). The ZnO-NPs treatment was found to be the most beneficial due to zinc’s essential role in supporting various biochemical and physiological processes, which was confirmed by proper physiological responses (Table 7) [20]. For other nutrients, including K, Mg, or Ca, no significant effects were found after NPs application. Our findings indicate relatively higher potassium content in seeds associated with SiO_2_-NPs-treatment that was also confirmed by Seleiman et al. [19] in potatoes. Similar to potatoes, sunflowers belong to the potassium preferring group of crops.

### 4.4. Response of Nanoparticles Spray Deposition to Quantity and Physiology of Sunflower

Several studies have reported positive results of foliar application of metal-based NPs on the growth and development of agriculturally attractive crops [27,80,81,82]. Furthermore, these studies have shown that the application of metal-based nanoparticles can support sustainable agriculture under inadequate long-term and seasonal conditions, such as salt stress [83], heavy metals contamination [84,85], or drought [86].

The most efficient treatments for sunflower growth and development, based on photosynthetic activity, chlorophyll content, transpiration rates (Table 7), stomata size, length and number (Appendix A), and water management, were found to be AuSi-NPs and ZnO-NPs, but also for sunflower development stages after first and second application in the case of AuSi-NPs treatment. This was partially unexpected because gold and silica are not typical plant nutrients [87], but their photocatalytic performance may be responsible for the encouraging physiology observed. These results are consistent with previous studies showing the photocatalytic properties of NPs in resistant nerve agent removal [40] and paracetamol degradation [88].

Monodisperse gold NPs with higher crystallinity and spherical morphology are considered to be highly efficient optical and electrical conductors, and capable of generating a significant amount of surface plasmons [35]. The exact mechanisms involved in energy transfer in plants are not yet fully understood, but it is believed to involve the stimulation of plant photochemical centers. Venzhik et al. [89] showed that gold nanoparticles concentration of 20 μg mL^−1^ can enhance photosynthesis, chloroplast content, and unsaturated fatty acid content in wheat, increase the number of thylakoids in grana, and improve tolerance to environmental stress in wheat. In addition, biogenic silicon dioxide with semiconductor properties encourages photosynthesis, photoprotection trough mechanisms associated with maximum photooxidation PSI, photochemical efficiency PSII systems, and photochemical gas exchange [90]. These effects may be reflected in leaf structures, such as the distribution, and width and length of trichomes and stomata (Table 3 and Table 4). Similarly, Thind et al. [70] described that foliar application of Si-NPs significantly improved photosynthetic pigments, flavonoid, TSP (total soluble protein), phenolic, FAA (free amino acid), proline, TSS (total soluble sugar), APX (ascorbate peroxidase), CAT (catalase), POD (peroxidase), and SOD (superoxide dismutase) enzyme activities of wheat.

Previous studies with ZnO-NPs spray deposition at lower concentrations have shown positive physiological responses in foxtail millet [30] and common sunflower [26]. Additionally, the intensification of chlorophyl production in maize (*Zea mays* L.) was observed in the ZnO-NPs-treatment [7]. Physiological response in ZnO and SiO_2_-NP_S_ treatments in our study corresponds to findings of Seleiman et al. [19] who observed enhanced potato yield-related parameters under water deficiency. It might be an important strategy in the near future, especially in arid regions worldwide, and applied NPs could mitigate the gradual climate changes.

Similarly, in our study, we observed improved physiological reactions after the second application during the flowering and seed development stages up to the ripening phase. This response was also reflected in the oil yield content (Table 4) and the food quality when considering linoleic acid as a fundamental human compound (Table 5).

In the case of Fe_3_O_4_-NPs treatment, only the index of stomatal conductance was statistically significant in contrast to NPs-free control (Table 7), where physiology has shown gradual trend at the stage of ripening (Figure 5c). This is a relatively surprising fact because iron is a micronutrient known to enhance growth and development, improve stress tolerance, and prevent chlorosis [21]. Furthermore, our hypothesis assumes that Fe_3_O_4_-NPs is an n-type semiconductor with magnetic properties and exhibit certain level of photocatalytical activity [91] that promotes analogical sunflower physiological responses. Herein, the insignificant physiology reactions could be an aspect of its stability, with preferred corrosion properties linked with changes in crystallinity; intensive solubility, mainly due to small dimension around 5 nm; and exposition to direct daily sunlight radiation; and variations of seasonal temperatures or precipitation (Figure 1). The stress reaction was manifested by a change of distribution, number of trichomes or modification of their physical parameters (Table 3), and by statistically insignificant quantitative parameters—head diameter, TSW, and grain yield. These results do not correspond well with Cai et al. [27] who, after Fe_3_O_4_-NPs application, observed prompt plant growth, generation of reactive oxide species (ROS) responsible for increased activity of antioxidant enzymes, level of salicylic acid synthesis, and expression of PR genes. Mahmoud et al. [72] stated that Fe-NPs significantly increased the content of macro-elements (N, P, K and Ca) in leaves of broad bean plants. Similar significance was observed in the micro-element content (Fe, Zn, and Mn) in broad bean seeds. The greatest concentration of Fe, Zn, and Mn in seed was observed in broad bean plants treated with Fe-NPs in comparison to control.

### 4.5. Impact of Nanoparticles Foliar Application on Agro-Ecological Community during Sunflower Vegetation Period Based on Determination of Epigeic Groups

Maintaining complex and diverse ecosystems is essential for the balanced populations of all integrated communities [92]. Our results according to collected epigeic insect populations (Table 8; Appendix A) showed the nonuniform trend in individual NPs treatments. Unexpectedly, the AuSi-NPs treatment and NPs-free control had a wider biodiversity of epigeic individuals, while Fe_3_O_4_-NPs and ZnO-NPs treatments negatively influenced the biodiversity of epigeic groups.

Despite the fact that Si-based materials are occasionally applied in sustainable agriculture [93,94,95], there were published works on their toxic effects on microorganisms [96,97], mainly in the nanoparticles range. However, our mesoporous biosilica is relatively accessible, not harmful to the soil environment [94], and did not negatively affect the developing stages of the epigeic communities, which are intimately tied to the soil system. Furthermore, the mesoporous biosilica was applied at a low concentration of around 10 mg.L^−1^, and most of it was adhered and absorbed by the leaf surfaces, with only a minor part gradually deposited onto the soil. In this context, Tian et al. [98] applied 100 mg L^−1^ of SiO_2_-NPs to *Brassica chinesis* L. by spraying and recorded alteration of soil metabolite profiles and a decline of microbial community.

The contribution of AuSi-NPs to insect biodiversity is interesting phenomena because we used lower content of Au NPs with biosilica and the most logical explanation could be appropriate agromanagement related to soil environmental condition. However, the role of ZnO-NPs treatment in the observed biodiversity is not clear, as this type of nanoparticle is commonly considered pro-environmental and even agronomically desired [26,30,39]. A similar tendency was shown with iron micronutrient NP treatment. In this case, the reason might be the higher dosage (76 mg L^−1^), small dimension and penetration into microorganisms [99,100], and relative solubility and reactivity of Fe_3_O_4_-NPs [101,102]. Trace content of residual citrate reagents should also be taken into consideration.

## 5. Conclusions

Our spray applied nanoparticles to sunflowers under field conditions resulted in beneficial responses within various quantitative, physiological and nutritional parameters in contrast to NPs-free control. Unique effects were observed for each of the applications, commercially available ZnO-NPs, biosynthetically formed Au-NPs fixed into mesoporous algae *Mallomonas kalinae* sp. nov. *(AuSi-treatment),* and synthetically prepared Fe_3_O_4_-NPs, all at relatively low concentration levels were used. We observed unexpected and beneficial responses in sunflower physiology, oil content, and even epigeic insect diversity.

The most surprising effects on sunflowers were observed in the AuSi-treatment where the number of leaf trichomes and the concentration of linoleic acid has increased. At the same time, an improvement was observed in quantitative parameters seasonal physiology, agronomically relevant oil yield content, and potential silica translocation to full ripe seeds. Most surprisingly, we observed a significant increase in agro-biological diversity based on epigeic insect groups, even though gold and silica are not essential nutrients for plants or insects.

Our previously published empirical findings conducted on several crops have shown that ZnO-NPs performed important roles in many agronomical aspects. This observation was also confirmed in the current experimental study in the case of quantitative parameters, yield, and plant physiology. This response was promptly stimulated after the second spray application, the most likely due to higher number of present leaves trichomes and stomata. From nutritional quality point of view, ZnO-NPs treatment induced the low content of human antinutrient—phosphorus—and the higher content of beneficial linoleic acid, but there was no beneficial translocation of zinc. The hypothesis about the expected wider agro-ecological diversity was not confirmed. On the contrary, in this treatment, the number of epigeic insect groups and species populations were the lowest, including NPs-free control.

Even the other micronutrient-based nanoparticles treatment with Fe_3_O_4_ did not show significant trends in trichome distribution at the lower number of leaf stomata, the quantitative and yield-related parameters, and oil content. Only full ripe seeds provided nutritional values of the lower antinutrient content, without iron translocation, but with beneficial concentration of linoleic acid. Regarding the physiological responses, only stomatal conductance reacted adequately at the statistically significant level, mostly at the ripening development stage. Additionally, agrobiological diversity of epigeic groups showed low abundance. The overall unconvincing role of Fe_3_O_4_-NPs treatment could be associated with very low dimension of iron nanoparticles around 5 nm, with great solubility and reactivity.

In case of control (NPs-free variant), we observed the significant number of stomata, nutritional content of essential fatty acid, such as oleic acid, and relatively abundant agroecological diversity, but with negative seasonal physiological response.

## Figures and Tables

**Figure 1 plants-12-01789-f001:**
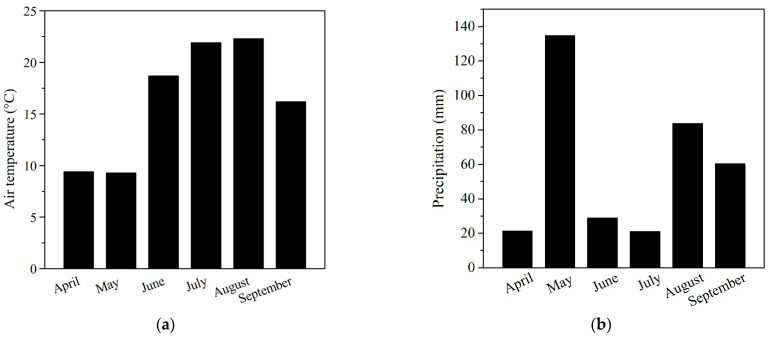
Monthly variations in (**a**) air temperature and (**b**) precipitation during the 2019 vegetation season at the experimental field Dolná Malanta in Nitra, Slovakia.

**Figure 2 plants-12-01789-f002:**
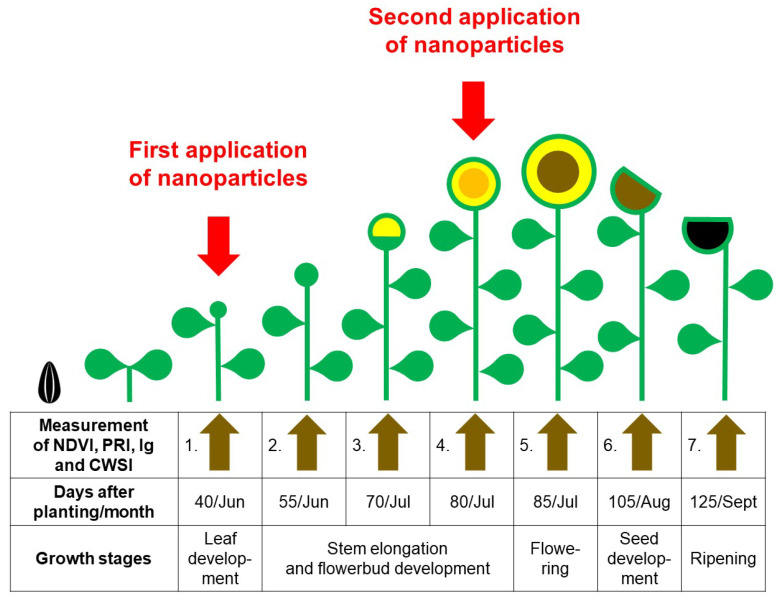
Schematic model of sunflower growth stages and application period of nanoparticles.

**Figure 3 plants-12-01789-f003:**
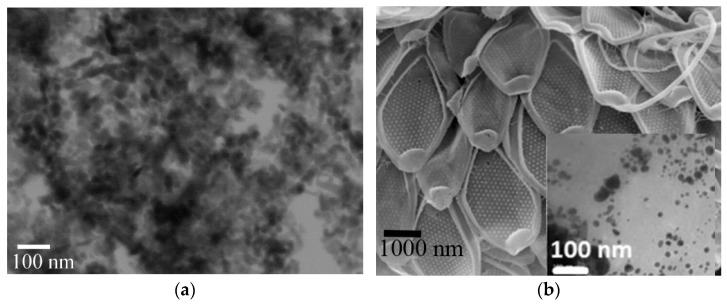
Treatments with three types of sprayed nanoparticles, (**a**) zinc oxide nanoparticles (SEM); (**b**) biosynthetically prepared Au NPs (inset) with mesoporous biosilica (algae *Mallomonas kalinae* sp. nov.) (AuSi-variant) (SEM); (**c**) morphology of Fe_3_O_4_-NPs with inset EDS analysis where main components were O and Fe corresponding to magnetite mineral, C is residual citrate part of synthesis procedure, and Cu resulted from Cu-grid; (**d**) X-ray diffraction analysis verifying the crystalline structure of magnetite.

**Figure 4 plants-12-01789-f004:**
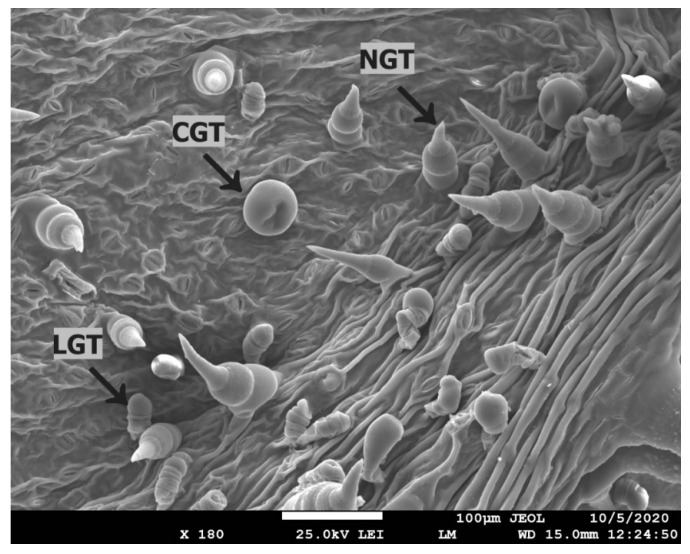
Trichome types on the abaxial side of sunflower leaf (NGTs—non-glandular trichomes, LGTs—linear glandular trichomes, CGTs—capitate glandular trichomes)—image made by SEM.

**Figure 5 plants-12-01789-f005:**
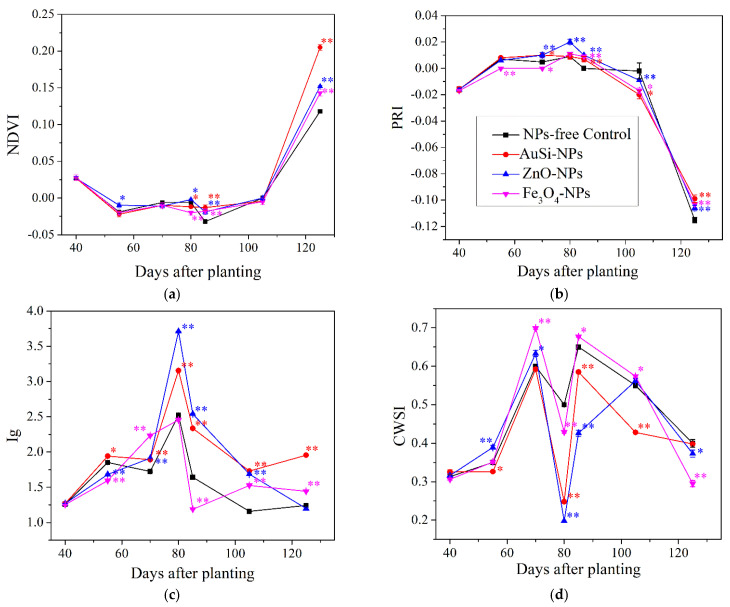
Evaluation of seasonal dynamics of foliar applied AuSi-NPs, ZnO-NPs, and Fe_3_O_4_-NPs treatments based on the physiological indexes in 2019 and compared with the NPs-free control; (**a**) normalized difference vegetation index (NDVI), (**b**) photochemical reflectance index (PRI), (**c**) Ig Stomatal conductance index (Ig), (**d**) crop water stress index (CWSI), the significance: * *p* value < 0.05, ** *p* value < 0.01.

**Table 1 plants-12-01789-t001:** Selected soil properties in autumn 2018 before sunflower planting at the experimental locality (Dolná Malanta).

pH	Carbonate Content (%)	Content of Humus (%)	Content of Nutrients (mg kg^−1^)				
			K	P	Na	Mg	Zn
6.49	0.18	2.23	425	63.8	65	331.6	1.96

**Table 2 plants-12-01789-t002:** Soil analysis to determine nitrogen fraction before sowing of sunflower in spring 2019.

Content of Nitrogen Forms in the Soil (mg kg^−1^)		
N Inorganic	N-NH_4_^+^	N-NO_3_^−^
14.8	7.0	7.8

**Table 3 plants-12-01789-t003:** Distribution of the trichomes (area = 1 mm^2^) on adaxial and abaxial side of leaves which were collected during flower bud formation after AuSi-NPs, Fe_3_O_4_-NPs, and ZnO-NPs foliar applications when compared with NPs-free control treatment (AD—adaxial, AB—abaxial, N—number of analyzed trichomes, NGTs—non-glandular trichomes, LGTs—linear glandular trichomes, CGTs—capitate glandular trichomes).

Type of Trichomes	Leaf Side	Control		AuSi-NPs		Fe_3_O_4_-NPs		ZnO-NPs	
		N	%	N	%	N	%	N	%
NGTs	AD	211	25.95	319	35.80	246	29.32	303	30.24
	AB	332	37.14	354	37.86	289	36.49	342	38.17
LGTs	AD	602	74.05	572	64.20	593	70.68	699	69.76
	AB	497	55.59	515	55.08	438	55.30	492	54.91
CGTs	AD	-	-	-	-	-	-	-	-
	AB	65	7.27	66	7.06	65	8.21	62	6.92
Summary	AD	813	100	891	100	839	100	1002	100
	AB	894	100	935	100	792	100	896	100

**Table 4 plants-12-01789-t004:** Comparison of quantitative and nutritional parameters of AuSi-NPs, ZnO-NPs, and Fe_3_O_4_-NPs treatments recorded in the 2019 vegetation season.

	AuSi- NPs	ZnO-NPs	Fe_3_O_4_-NPs	Control
Quantitative Parameters				
Number of Plants per Hectare (pcs)	78,123 ± 1798 ^ns^	77,659 ± 2217 ^ns^	77,291 ± 1565 ^ns^	77,982 ± 2007
Number of Heads per Hectare (pcs)	78,123 ± 1798 ^ns^	77,659 ± 2217 ^ns^	77,647 ± 2162 ^ns^	78,249 ± 2133
Head Diameter (mm)	245 ± 6 **	220 ± 17 *	198 ± 23 ^ns^	172 ± 13
Weight of Dry Seed Head (g)	186.11 ± 16.22 **	172.84 ± 7 **	166.28 ± 21.20 *	109.64 ± 12.39
Weight of Thousand Seeds (g)	57.43 ± 0.29 **	59.98 ± 0,15 **	59.19 ± 0.46 ^ns^	59.35 ± 0.17
Grain Yield (t ha^−1^)	2.75 ± 0.11 ^ns^	3.29 ± 0.18 **	2.71 ± 0.13 ^ns^	2.47 ± 0.17
Nutritional Parameter				
Content of Oil (%)	47.50 ± 3.28 ^ns^	50.02 ± 2.47 ^ns^	47.68 ± 3.80 ^ns^	52.24 ± 1.27

Note: The significance: * *p* value < 0.05, ** *p* value < 0.01, ^ns^ Non-significant. The values present after symbol ± show the variance of the values as standard deviation.

**Table 5 plants-12-01789-t005:** Final quality of sunflower oil based on the content of fatty acids after nanoparticles treatment including ZnO-NPs, AuSi-NPs, and Fe_3_O_4_-NPs variants in comparison with a NPs-free control.

Fatty Acids in Sunflower Oil (%)	AuSi-NPs	ZnO-NPs	Fe_3_O_4_-NPs	Control
Oleic Acid	34.04 ± 0.15 **	33.74 ± 0.65 **	33.77 ± 0.24 **	37.24 ± 0.99
Linoleic Acid	54.37 ± 0.28 **	54.86 ± 0.39 **	54.65 ± 0.11 **	51.57 ± 0.3
Palmitic Acid	5.94 ± 0.31 ^ns^	5.85 ± 0.03 ^ns^	5.91 ± 0.85 ^ns^	5.60 ± 0.72
Stearic Acid	3.68 ± 0.33 ^ns^	3.62 ± 0.18 ^ns^	3.70 ± 0.20 ^ns^	3.63 ± 0.21
Arachidic Acid	0.2564 ± 0.0040 ^ns^	0.2480 ± 0.0042 ^ns^	0.2562 ± 0.0031 ^ns^	0.2494 ± 0.0037 ^ns^
cis-11-eicosenoic Acid	0.1485 ± 0.0054 ^ns^	0.1501 ± 0.0060 ^ns^	0.1491 ± 0.0049 ^ns^	0.1605 ± 0.0056 ^ns^
Behenic Acid	0.8135 ± 0.0112 ^ns^	0.7988 ± 0.0100 ^ns^	0.8210 ± 0.0116 ^ns^	0.8022 ± 0.0120 ^ns^
Lignoceric Acid	0.2299 ± 0.0075 ^ns^	0.2228 ± 0.0080 ^ns^	0.2259 ± 0.0068 ^ns^	0.2385 ± 0.0073 ^ns^

Note: C16:0—palmitic acid, C16:1—palmitoleic acid, C18:0—stearic acid, C18—1cis n9-oleic acid, C18:2—cis n6-linolenic acid, C18:3, n3—α-linolenic acid, C20:0-arachidic acid, C20:1 n9-cis-11-eicosenoic acid, C22:0—behenic acid; The significance: ** *p* value < 0.01, ^ns^: non-significant. The values present after symbol ± show the variance of the values as standard deviation.

**Table 6 plants-12-01789-t006:** Chemical composition analysis of sunflower fully ripe seeds with various nanoparticles treats, variants such as ZnO-NPs, AuSi-NPs, Fe_3_O_4_-NPs, and NPs-free control.

	AuSi-NPs	ZnO-NPs	Fe_3_O_4_-NPs	Control
Zinc	NA	0.058 ± 0.005	NA	0.073 ± 0.007
Iron	0.047 ± 0.003	0.041 ± 0.004	0.040 ± 0.002	0.042 ± 0.001
Silica	0.050 ± 0.015	NA	NA	0.025 ± 0.009
Phosphor	8.82 ± 0.441	7.76 ± 0.388	8.35 ± 0.42	9.07 ± 0.45
Potassium	7.19 ± 0.66	5.90 ± 0.90	6.93 ± 0.45	6.89 ± 0.54
Calcium	1.30 ± 0.02	0.95 ± 0.09	1.14 ± 0.09	1.36 ± 0.50
Magnesium	3.64 ± 0.44	3.38 ± 0.36	3.48 ± 0.25	3.80 ± 0.20

Note: NA—not analyzed. The values present after symbol ± show the variance of the values as standard deviation.

**Table 7 plants-12-01789-t007:** List of AuSi-NPs-, ZnO-NPs-, and Fe_3_O_4_-NPs-treated sunflower plants physiological parameters in comparison with NPs-free control.

Parameter	AuSi-NPs	ZnO-NPs	Fe_3_O_4_-NPs	Control
NDVI	0.025 ± 0.003 **	0.02 ± 0.003 **	0.014 ± 0.004 ^ns^	0.012 ± 0.001
PRI	−0.014 ± 0.0017 *	−0.013 ± 0.001 *	−0.016 ± 0.001 ^ns^	−0.019 ± 0.002
Ig	2.04 ± 0.006 **	2 ± 0.003 **	1.67 ± 0.003 **	1.628 ± 0.005
CWSI	0.414 ± 0.002 **	0.414 ± 0.005 **	0.476 ± 0.004 ^ns^	0.48 ± 0.006

Note: NDVI—normalized difference vegetation index, PRI—photochemical reflectance index, Ig—stomatal conductance index, CWSI—crop water stress index. The significance: * *p* value < 0.05, ** *p* value < 0.01, ^ns^: non-significant. The values present after symbol ± show the variance of the values as standard deviation.

**Table 8 plants-12-01789-t008:** Abundance and dominance of epigeic groups in the studied treatments with common sunflower during vegetation season 2019.

Epigeic Groups	Control	AuSi-NPs	ZnO-NPs	Fe_3_O_4_-NPs	Σ	Dominance (%)
Acarina	122	191	92	148	553	11.67
Aphidoidae	ND	ND	ND	3	3	0.06
Araneida	36	60	34	83	213	4.49
Auchenorrhyncha	ND	1	ND	1	2	0.04
Coleoptera	766	789	266	282	2103	44.38
Collembola	164	342	255	183	944	19.93
Dermaptera	6	6	2	5	19	0.40
Diplopoda	5	2	6	2	15	0.33
Diptera	15	23	12	6	56	1.19
Formicoidae	52	101	87	81	321	6.77
Heteroptera	4	8	2	7	21	0.44
Hymenoptera	ND	9	3	4	31	0.65
Chilopoda	5	13	4	5	27	0.57
Larvae	10	35	19	20	84	1.77
Lepidoptera	1	ND	ND	ND	1	0.02
Lumbricida	2	2	ND	3	7	0.16
Muridae	-	1	ND	1	2	0.04
Opilionida	35	78	44	46	203	4.29
Orthoptera	24	34	40	33	131	2.76
Siphonaptera	ND	2	ND	ND	2	0.04
Σ	1262	1697	866	913	4738	100.00

Note: ND—not detected.

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
