# Peer review of "Agronomic Investigation of Spray Dispersion of Metal-Based Nanoparticles on Sunflowers in Real-World Environments"

_plants, 2023, doi:10.3390/plants12091789_

Round 1

Reviewer 1 Report

Comments on plants-2288204

Abstract

There should be a presentation of results quantitatively like an increase or decrease in various parameters

Introduction

The whole manuscript needs to be checked regarding various typos and grammatical mistakes e.g. “Figure 2. Figure 1. Monthly variations”, “Analyzation of chemical elements”, and many others

The role of NPs in abiotic stress tolerance may be highlighted by consulting the following paper: https://doi.org/10.1007/s12517-021-07384-w

Materials and Methods

The main issue with this manuscript is the duration of research work under field conditions. Field experiments could only be reliable if these are conducted for s=consecutively two years

“Application was performed early in the morning in wind-less conditions until the sunflower leaves were completely wet.” It is quite confusing regarding the exact amount applied, please clarify the amount applied in both sprays

On what basis doses of different NPs were calculated and applied? Was there a toxicological study conducted regarding the toxicity of various levels of different NPs tested?

The authors claimed, “Biosynthetically prepared AuSi-NPs and ZnO-NPs”. I am unable to understand how these NPs were biosynthesized?

All the tables should be formatted properly following the same style in all

In each Table, the values present after ± should be explained as a footnote of the Table

The number of Tables may be reduced by presenting only the main parameters supporting the hypothesis and the rest as supplementary material

The introduction and discussion need to be updated with recent papers from 2023. The following papers may be consulted and cited

Tanveer, Y., Jahangir, S., Shah, Z.A., Yasmin, H., Nosheen, A., Hassan, M.N., Illyas, N., Bajguz, A., El-Sheikh, M.A. and Ahmad, P., 2023. Zinc oxide nanoparticles mediated biostimulant impact on cadmium detoxification and in silico analysis of zinc oxide-cadmium networks in Zea mays L. regulome. Environmental Pollution, 316, p.120641.

Faizan, M., Alam, P., Rajput, V.D., Faraz, A., Afzal, S., Ahmed, S.M., Yu, F.Y., Minkina, T. and Hayat, S., 2023. Nanoparticle Mediated Plant Tolerance to Heavy Metal Stress: What We Know?. Sustainability, 15(2), p.1446.

Giri, V.P., Shukla, P., Tripathi, A., Verma, P., Kumar, N., Pandey, S., Dimkpa, C.O. and Mishra, A., 2023. A Review of Sustainable Use of Biogenic Nanoscale Agro-Materials to Enhance Stress Tolerance and Nutritional Value of Plants. Plants, 12(4), p.815.

Seleiman, M.F., Al-Selwey, W.A., Ibrahim, A.A., Shady, M. and Alsadon, A.A., 2023. Foliar applications of ZnO and SiO2 nanoparticles mitigate water deficit and enhance potato yield and quality traits. Agronomy, 13(2), p.466.

Zahedi, S.M., Hosseini, M.S., Karimi, M., Gholami, R., Amini, M., Abdelrahman, M. and Tran, L.S.P., 2023. Chitosan-based Schiff base-metal (Fe, Cu, and Zn) complexes mitigate the negative consequences of drought stress on pomegranate fruits. Plant Physiology and Biochemistry.

Author Response

Dear reviewer, thank you for your revision and helpful suggestions that contributed to the quality of our manuscript. Our responses to your comments are enclosed as a cover letter. Thank you once again.

Reviewer 2 Report

Nanoparticles are of great interest due to their unique properties. They are widely used in many fields of science and technology.

  This work is devoted to the study of the agronomic effect of spraying nanoparticles on sunflower. The following nanoparticles are used: gold nanoparticles anchored to SiO2 mesoporous silica (AuSi-NPs), zinc oxide nanoparticles (ZnO-NPs) and iron oxide nanoparticles (Fe3O4 NPs). It was shown that biosynthetically produced AuSi NPs and ZnO NPs were highly effective in improving the seasonal physiology of sunflower, increasing the spread of trichomes on leaves, and increasing oil yield per hectare. The control with AuSi-NP and no NP showed the widest insect biodiversity, while the ZnO-NP treatment showed the lowest phosphorus content as an anti-nutrient. 

There are a number of comments and questions

   The work contains references to works where the parameters of nanoparticles are indicated (Materials and methods). It is better to write the dimensions and characteristics in the text of the article.

- A typo in the caption to the figure. Written: Figure 2 Figure 1

- No errors on some charts and tables

- What is the choice of nanoparticles and concentration related to?

Author Response

(The authors gave the same response as above.)

Round 2

Reviewer 1 Report

All the comments have been well incorporated

The references in a research article should be around 60. However, there are 100+ references cited in the manuscript. The authors are highly suggested to reduce 

Correct the unit of temperature in Figure 1a

Author Response

Dear reviewer, thank you for your comments and suggestions. Our responses are attached in cover letter.
